

# Microbial fungicides can positively affect aubergine photosynthetic properties, soil enzyme activity and microbial community structure

Longxue Wei, Jinying Zhu, Dongbo Zhao, Yanting Pei, Lianghai Guo, Jianjun Guo, Zhihui Guo, Huini Cui, Yongjun Li and Jiansheng Gao

Dezhou Institute of Agricultural Science, Dezhou, Shandong, China

## ABSTRACT

**Background**. This study examined the effects of microbial agents on the enzyme activity, microbial community construction and potential functions of inter-root soil of aubergine (*Fragaria* × *ananassa* Duch.). This study also sought to clarify the adaptability of inter-root microorganisms to environmental factors to provide a theoretical basis for the stability of the microbiology of inter-root soil of aubergine and for the ecological preservation of farmland soil.

**Methods**. Eggplant inter-root soils treated with *Bacillus subtilis* (QZ_T1), *Bacillus subtilis* (QZ_T2), *Bacillus amyloliquefaciens* (QZ_T3), *Verticillium thuringiensis* (QZ_T4) and *Verticillium purpureum* (QZ_T5) were used to analyse the effects of different microbial agents on the inter-root soils of aubergine compared to the untreated control group (QZ_CK). The effects of different microbial agents on the characteristics and functions of inter-root soil microbial communities were analysed using 16S rRNA and ITS (internal transcribed spacer region) high-throughput sequencing techniques.

**Results**. The bacterial diversity index and fungal diversity index of the aubergine inter-root soil increased significantly with the application of microbial fungicides; gas exchange parameters and soil enzyme activities also increased. The structural and functional composition of the bacterial and fungal communities in the aubergine inter-root soil changed after fungicide treatment compared to the control, with a decrease in the abundance of phytopathogenic fungi and an increase in the abundance of beneficial fungi in the soil. Enhancement of key community functions, reduction of pathogenic fungi, modulation of environmental factors and improved functional stability of microbial communities were important factors contributing to the microbial stability of fungicide-treated aubergine inter-root soils.

## INTRODUCTION

Soil microorganisms play a crucial role in maintaining soil function and inter-root ecosystem sustainability and are often considered to be sensitive bioindicators of soil health (*Wu et al., 2024*). The structure and distribution of microbial communities are

Corresponding author
Jiansheng Gao, gaojian-sheng1990@163.com

influenced by a variety of both biotic and abiotic factors (*e.g.*, apoplastic inputs, soil physicochemical properties, soil nutrient status, and changes in plant diversity). These factors control changes in microbial community structure and diversity (*Edwards et al., 2015*). Understanding the ecological processes involved in microbial community assembly helps elucidate how microbial community composition responds to environmental change, where deterministic and stochastic processes are based on ecological niche and neutrality theories, which are commonly used to explain microbial community construction (*Ning et al., 2019*). Deterministic processes determine the presence or absence and relative abundance of species and are associated with ecological selection (*Guo et al., 2022*). Stochastic processes include unpredictable perturbations, probabilistic dispersal, and random birth-death events. Deterministic and stochastic processes act simultaneously to regulate the construction of ecological communities (*Chen et al., 2021*).

Microbials are new types of fertilisers made with one or more functional strains of bacteria, fermented at high density (*Seenivasagan & Babalola, 2021*). The need to control pests and diseases without affecting the environment and the increased demand for sustainable crop yields are driving the global market for soil microbials. Microbial agents mainly based on bacteria and fungi are applied to the soil as an alternative to traditional inorganic fertilizers (biofertilizers) to perform specific functions including biological control of pests and diseases (*Ji et al., 2023*), bioremediation and soil improvement (*Pankaj & Nandita, 2021*). Although some microbial fungicides (*e.g.*, rhizobacteria) have a long and fruitful history of use (*Freiberg et al., 1997*), the performance of fungicides in the field is not consistent (*Díaz-pérez, 2022*). *Bacillus spp.*, being a rhizospheric bacteria, has the potential to provide protection from biotic stresses and also to enhance plant growth (*Chandra et al., 2024*) due to its ability to colonise plant roots in the presence of competing *Pseudomonas* bacteria (*Pomerleau et al., 2024*). Previous studies have shown the up-regulation of plant defence enzymes, such as peroxidase (POX), superoxide dismutase (SOD) and catalase (CAT), and the reduction of cellular stress indicators after the addition of *Trichoderma harzianum*, which also activated the defence effect of tomato seedlings and significantly protected tomato plants against early blight (*Khalil & Youssef, 2024*; *Prigigallo et al., 2023*). Biocontrol fungicides are a desirable strategy for improving control of soil-borne pathogens: *Trichoderma harzianum* has been shown to have a high biocontrol capacity and to improve yield (*Núñez Palenius Héctor et al., 2022*), *Bacillus amyloliquefaciens* and *Purpureocillium lilacinum* can enhance repellency and root-knot nematode control (*DePaula et al., 2024*), and *Paecilomyces lilacinus* and *Verticillium chlamydosporium* have antagonistic effects on southern root-knot and cyst nematodes (*Nyongesa, Coosemans & Kimenju, 2007*; *Dackman & Bååth, 2018*).

A more accurate understanding of the ecological functions and modes of action of mycorrhizal fungicides is key to optimising their efficacy and guiding their targeted use in crop production (*O'Callaghan, Ballard & Wright, 2022*). Fewer studies have been conducted to characterise the inter-root soil microbial community of aubergine under different microbial fungicide treatments and their functional predictions, which are crucial for predicting the role of inter-root microbes in regulating the functioning of plant-microbe ecosystems. This study investigated the effects of microbial agents on

the photosynthetic characteristics, soil enzyme activities and soil microbial community structure composition of aubergine under different microbial agents, and analysed the changes of soil fungal nutrient phenotypes and bacterial community functions under microbial agent treatments with FUNGuild and FAPROTAX, respectively, using 16S rRNA and ITS high-throughput sequencing technologies. The results of this study can help reveal the soil ecological stabilisation mechanism of microbial fungicide in promoting growth and disease resistance, and evaluate the practical application effect of different carrier fungicides in agricultural production, providing theoretical support for the sustainable development of the aubergine industry.

## MATERIALS & METHODS

### Test materials and test site

This experiment was conducted from January 19, 2023 to February 17, 2023 at Xin Sheng Seed Industry, Pingyuan County, Dezhou City, and from February 19, 2023 to September 2023 at the greenhouse of Dazhuang Village, En Township, Pingyuan County, Dezhou City (120.4114°E, 30.4406°N), with aubergine varieties of "Big Red Robe" as the test crop. aubergine was the previous crop in the test site. The basic physicochemical properties of the soil were organic matter 22.88 g $kg^{-1}$, total nitrogen 1.51 g $kg^{-1}$, alkaline nitrogen 144.64 mg $kg^{-1}$, effective phosphorus 115.45 mg $kg^{-1}$, and fast-acting potassium 120.13 mg $kg^{-1}$.

The following microbial agents were used for testing: QZ_T1: *Bacillus subtilis* (Biovox Razen, wettable powder, an effective live bacteria count of $1.00 \times 10^9$ CFU $mL^{-1}$.); QZ_ T2: *Trichoderma harzianum* (Biovox Rootshield, wettable powder, an effective live bacteria count of $3.00 \times 10^7$ CFU $mL^{-1}$); QZ_T3: *Bacillus amyloliquefaciens* (Shanxi Xiannong Biotechnology Co. Ltd. Xiannong Root Shield, wettable powder, an effective live bacteria count of $1.00 \times 10^9$ CFU $mL^{-1}$); QZ_T4: *verticillium chlamydosporium* sp. (Yunnan Microstructure Source, micronutrient, an effective live bacteria count of $2.50 \times 108$ CFU $mL^{-1}$); QZ_T5: *Paecilomyces lilacinus* (Chongqing micro-nuclear biological purple warrior, wettable powder, an effective live bacteria count of $1.00 \times 10^8$ CFU $mL^{-1}$) (Table 1). All five microbial fungicides were purchased from the biological plant protection station of the Fruit Yau Agricultural Resources and Beihai Qunlin biological factory store. The test was conducted in a plastic greenhouse with double-layer thermal insulation. A control plot (QZ_CK) and five microbial fungicide treatment plots (QZ_T1, QZ_T2, QZ_T3, QZ_T4, QZ_T5) were cultivated in the greenhouse. All treatments were fertilized according to the routine application of fertilizers: $1.5 \times 10^4$ kg $hm^{-2}$ of rice husk, $2.25 \times 10^4$ kg $hm^{-2}$ of cow dung, $1.2 \times 10^3$ kg $hm^{-2}$ of compound fertilizer (purchased from the Agricultural Store in Pingyuan County, Texas, with the composition of 17-17-17 N-$P_2O_5$-$K_2O$), $1.2 \times 10^3$ kg $hm^{-2}$ of calcium fertilizer and $1.2 \times 10^3$ kg $hm^{-2}$ of diammonium ammonium. Soil fertilization was followed by flooding and smothering, and then the planting of eggplant seedlings. Fruit expansion fertilizer was used three times in the later stages (15-5-30) with 2 kg of fertilizer used each time. Eggplant seedlings were planted on March 20, and inter-root furrow application of five fungicides (dosage of 150 times the
**Table 1  Experimental treatments.**

| Samples | Treatments |
|---------|------------|
| QZ_CK | No fungicide applied |
| QZ_T1 | *Bacillus subtilis* |
| QZ_T2 | *Trichoderma harzianum* |
| QZ_T3 | *Bacillus amyloliquefaciens* |
| QZ_T4 | *Verticillium chlamydosporium* |
| QZ_T5 | *Paecilomyces lilacinus* |

liquid, each time 100 mL per plant) occurred on April 20 and May 19. The control plot was managed with conventional methods and no fungicides. Plots were randomly assigned and the experimental plot area of each treatment was 50 m$^2$, with three replications, and a plant spacing of 60 cm $\times$ 80 cm.

## Measurement of photosynthetic parameters of eggplant

The photosynthetic parameters of the plants (middle and upper inverted three leaves) were determined on May 31, 2023 at 10:00 am using a CIRAS-4 portable photosynthetic determination system (Li-Cor, Lincoln, NE, USA), with an LED red and blue light source leaf chamber (90 $\mu$molmm$^2$ s$^{-1}$s for red light and 10 $\mu$molmm$^2$ s$^{-1}$ for blue light), a leaf chamber area (Aleaf) of 1.75 cm$^{-2}$, a leaf chamber temperature (Tcuv) of 28.80 $\pm$0.85 °C, a leaf temperature (Tleaf) of 28.92 $\pm$1.11 °C, and a leaf chamber relative humidity (RH) of 59.54% $\pm$5.23%. The reference $CO_2$ concentration was controlled at 450.19 $\pm$3.31 $\mu$mol mol$^{-1}$, the vapor pressure deficit was 0.9 $\pm$0.39 mb, and the light intensity within the leaf chamber was set at 1400 $\mu$mol m$^{-2}$ s$^{-1}$.

## Determination of soil physical and chemical properties

On June 26, 2023, the soil was collected from each plot at a depth of 0 to 20 cm using a five-point sampling method and a soil auger. The samples were mixed well, then plant residues, gravel and other debris in the soil samples were removed. Part of the sample was stored in the refrigerator at $-80$ °C for determining soil microorganisms and enzyme activities, and part of it was air-dried and passed through a two mm sieve for determining soil physicochemical properties. Soil organic matter content was titrated using the redox method after high temperature digestion. Total nitrogen content was titrated by an acid standard solution using the semi-trace Kjeldahl method. Alkali nitrogen content was titrated by an acid standard solution using the alkali diffusion method. Available phosphorus content was extracted by 0.5 mol L$^{-1}$ NaHCO$_3$ and then determined by antimony using the molybdenum colorimetric method, and then detected by a TU-1810 UV-Vis Spectrophotometer (Beijing Pudian General Instrument Co., Ltd., Beijing, China). Quick-acting potassium (K) content was detected by a TU-1810 UV-Vis Spectrophotometer. The content of available potassium (AK) was detected by CH$_3$COONH$_4$ extraction and then detected by a BWB-1 flame spectrophotometer (BWB Technologies, Yorba Linda, CA, USA).

## Determination of soil enzyme activities

Soil sucrase was determined by the colorimetric method of 3,5-dinitrosalicylic acid; soil urease was determined by the colorimetric method of sodium phenol sodium hypochlorite; soil catalase was determined by the titrimetric method of potassium permanganate; alkaline phosphatase was determined by the sodium benzene phosphate colorimetric method. All the steps were performed by the China Rice Research Institute according to the instructions of the corresponding enzyme activity kits provided by Suzhou Keming Biotechnology Co. Ltd, and the samples were colorimetrically determined using a multifunctional enzyme marker (TECAN-Spark 20M).

## Analysis of microbial diversity measurements of inter-root soils of aubergines

Soil samples were collected on 26 June 2023 for microbial diversity determination. Aubergine plants were collected together with the surrounding soil using a spade, keeping the root system intact as much as possible. The peripheral soil of the root system was then shaken off the roots and the soil immediately adjacent to the aubergine root system was gently brushed with a sterile brush to remove plant stubs, sand, gravel, mulch, earthworms and other foreign materials. Five aubergine inter-root soils were randomly selected, mixed well and then placed into sterile plastic bags. Three samples were collected from each group. Finally, the soil samples were transported to the laboratory and stored in the refrigerator at $-80\,^{\circ}C$ in an insulated bucket with ice.

High-throughput sequencing analyses were performed by Shanghai Meiji Biomedical Technology Co. Ltd. using the labelled bacterial universal primer 338F (5′-ACTCCTACGGGGAGGCAGCA-3′)-806R (5′-GGACTACHVGGGTWTCTAAT-3′) and the fungal universal primer ITS1F (5′-CTTGGTCATTTAGAGGAAGTAA-3′)-ITS2R (5′-GCTGCGTTCTTCATCGATGC-3′) primer was used to amplify the soil bacterial 16SrRNA gene region and the soil fungal ITS gene region. Afterwards, the raw data were quality-filtered and merged on the Illumina MiSeq platform (Illumina, San Diego, CA, USA) according to the standard method provided by Majorbio (Shanghai, China). The raw data were then clustered at a 97% similarity level to obtain the OTU (operational taxonomic unit). Species annotation was performed on the OTU sequences, and the community composition, alpha diversity and relative abundance of the samples were counted at each taxonomic level. Heatmap, redundancy and ranked regression analyses were performed within the platform on the correlation between the soil environmental factors and microbial communities to determine the effects of soil environmental factors on the composition of microbial communities.

## Data processing and analysis of high-throughput sequencing results

Trimmomatic software was used for quality control of the raw sequences, FLASH1.2.11 software was used for double-ended sequence splicing, and UPARSE7.1 software was used for OTU clustering of the sequences based on 97% similarity. Based on the OTU clustering results, the Majorbio Cloud platform was used for further data analysis and information mining, including species community composition analysis, significance test of intergroup

differences and diversity analyses (including Sobs index, Simpson index, Shannon-Wiener index, Chao1 richness and abundance-based coverage estimator (ACE)); 6S gene function prediction analysis was also performed using the COG database.

The data were statistically analysed using Excel 2007 and SPS Statistics 20.0 software, and the soil microbial diversity analysis and visual mapping were performed using Origin2019b and the MajorBIO cloud platform. The OTUs were subjected to multiple sequence comparison and combined with the environmental factors to perform the redundancy analysis (RDA) to obtain the influence group (OG), the effect group (OG) and the environmental influences affecting the community changes between groups. The results were plotted using Canoco 5.0. Analysis of variance (ANOVA) and multiple comparisons ($P < 0.05$) were performed using one-way ANOVA and the least significant difference (LSD) method.

## RESULTS

### Effects of different microbial agents on photosynthetic characteristics of aubergine

As shown in Table 2, the application of different microbial agents had different effects on the net photosynthetic rate, stomatal conductance, intercellular $CO_2$ concentration and transpiration rate parameters of aubergine leaves. There were significant differences between the five fungicide treatments and the control (QZ_CK) in the four photosynthetic parameters of net photosynthetic rate, stomatal conductance, inter-cellular carbon dioxide concentration and transpiration rate, with the QZ_T2 treatment having the highest net photosynthetic rate, stomatal conductance, and transpiration rate, and the lowest inter-cellular carbon dioxide. The differences between treatments and the control were not significant for the water utilisation of the leaves, except between the control (QZ_CK) and the QZ_T4 treatment. This indicates that the application of microbial fungi is beneficial to the enhancement of photosynthetic characteristics of aubergine leaves, and the application effects of microbial fungi of different compositions are different.

### Effect of different microbial agents on enzyme activities of aubergine soil

The application of different microbial agents had different effects on soil sucrase, alkaline phosphatase, catalase and urease activities (Fig. 1).

In the comparison of soil sucrase activity of the five fungicide treatment combinations and the control, only the difference between the QZ_T2 treatment and the control (QZ_CK) reached a significant level; the QZ_T2 treatment had the highest sucrase activity, which was 67.48% higher than that of QZ_CK (Fig. 1A). There was a significant difference in the alkaline phosphatase activity of the QZ_T5, QZ_T4, QZ_T1 and QZ_T2 treatments compared to the control (QZ_CK), and there was a significant difference in alkaline phosphatase activity of four treatments compared to the control, with phosphatase activity levels 19.87% (QZ_T1), 17.88% (QZ_T2), 20.75% (QZ_T4) and 28.54% (QZ_T5) higher than QZ_CK, respectively; the alkaline phosphatase activity of QZ_T3 was 4.56% higher than the control QZ_CK, but the difference was not significant (Fig. 1B). Three

**Table 2  Photosynthetic characteristics of eggplant leaves.**

| Treatment | Net photosynthetictate μmol m$^{-2}$ s$^{-1}$ | Stomatal conductance mmol m$^{-2}$ s$^{-1}$ | Inter cellular CO$_2$ concentration umol mol$^{-1}$ | Transpiration rate mmol m$^{-2}$ s$^{-1}$ | Water use efficiency mmol mol$^{-1}$ |
|---|---|---|---|---|---|
| QZ_CK | 22.80 ± 0.20[c] | 0.200 ± 0.010[c] | 470.03 ± 4.04[a] | 8.93 ± 0.04[d] | 2.55 ± 0.02[c] |
| QZ_T1 | 23.91 ± 0.07[b] | 0.333 ± 0.015[a] | 408.97 ± 12.89[c] | 9.10 ± 0.16[c] | 2.63 ± 0.05[b] |
| QZ_T2 | 26.12 ± 0.03[a] | 0.373 ± 0.031[a] | 407.37 ± 5.40[c] | 9.37 ± 0.04[a] | 2.79 ± 0.01[a] |
| QZ_T3 | 24.07 ± 0.03[b] | 0.283 ± 0.032[b] | 431.80 ± 10.07[b] | 9.21 ± 0.05[bc] | 2.61 ± 0.02[b] |
| QZ_T4 | 23.98 ± 0.08[b] | 0.299 ± 0.045[b] | 418.75 ± 8.25[b] | 9.25 ± 0.09[abc] | 2.59 ± 0.02[bc] |
| QZ_T5 | 25.97 ± 0.16[a] | 0.360 ± 0.036[a] | 408.67 ± 7.72[c] | 9.29 ± 0.05[ab] | 2.80 ± 0.03[a] |

**Notes.**

Pn, Net photosynthetic rate; Gs, Stomatal conductance; Ci, Intercellular CO$_2$ concentration; Tr, Transpiration rate; WUE, Water use efficiency.

The data in the table are the mean±standard error ($n = 3$).

Different lowercase letters indicate significant differences at the 0.05 level between different treatments.

treatments, QZ_T1, QZ_T2 and QZ_T5, had significantly different levels of catalase activity compared with the control treatment, at levels 6.83%, 8.60%, and 3.41% higher than QZ_CK, respectively (Fig. 1C). Urease activity differed significantly between all five bacterial treatments and the control treatment, with levels 43.48%, 46.41%, 44.97%, 62.38% and 35.32% higher than QZ_CK, respectively (Fig. 1D). The different effects of different microbial agents on the soil enzyme activity of aubergine may be due to different mechanisms of action and the interaction of soil physicochemical properties on enzyme activity.

## Sample sequencing depth assessment and OTU cluster analysis

The dilution curve reached the plateau period, indicating that the sequencing data reflected the microbial community diversity and that the dilution curve also indirectly reflected the species richness. Sequencing of fungal samples yielded a total of 360,793,91,218,163 bases (Fig. 2A), with an average sequence length of 252 bp, while sequencing of bacterial samples yielded a total of 322,253,134,652,822 bases, with an average sequence length of 417 bp (Fig. 2B); OTU sequence similarity was 0.97. USEARCHv11 in UPARSE-OTU was used for the sequencing depth and OTU clustering analyses. The dilution curves produced by UPARSE-OTU were gradually flattened, indicating that the sequenced.

## Alpha diversity of soil bacteria and fungi after treatment with different microbial agents

Alpha diversity refers to the number of species and species diversity in localised habitats or ecosystems, and is mainly analysed using Sobs index, Simpson index, Shannon-Wiener index, Chao1 richness, and abundance-based coverage estimator (ACE). As shown in Table 3, the coverage of soil fungi samples exceeded 99%, and the coverage of bacterial samples reached 98%, indicating that the results of this study reflected the diversity of microbial communities in the samples and that the sequencing results were reasonable. The Sobs index, Simpson index and Shannon-Wiener index results showed no significant differences in the soil fungal and bacterial communities in the inter-root soil of aubergine in

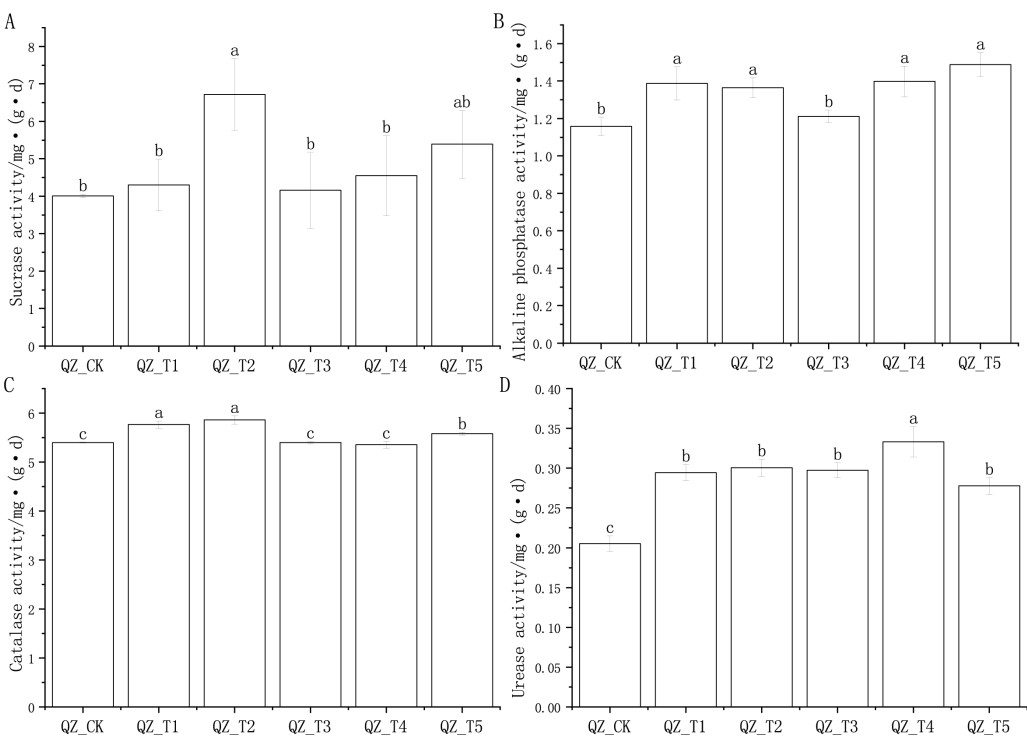

**Figure 1   Effects of different microbial agents on soil enzyme activities in the rhizosphere of eggplant.**
Note: Different lowercase letters indicate significant differences at the 0.05 level between different treatments.

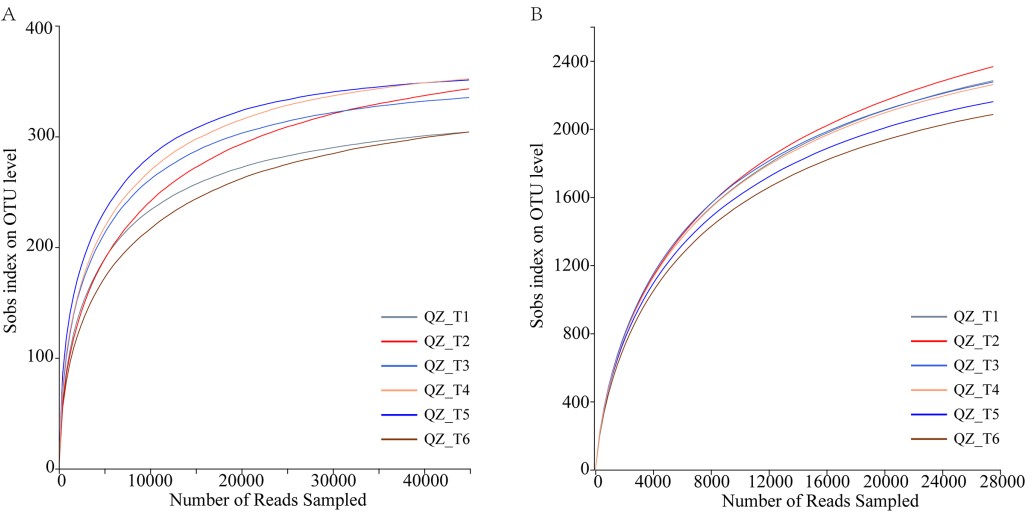

**Figure 2   Microbial diversity rarefaction curves.**

**Table 3  Alpha diversity index of microbial community in rhizosphere soil of eggplant.**

| Kingdom | Sample | Abundance-based coverage estimator | Chao1 richness | Shannon-Wiener index | Simpson index | Sobs index | Coverage % |
|---|---|---|---|---|---|---|---|
| Fungi | QZ_CK | 318.21 ±10.19[c] | 321.40 ±19.37[c] | 3.05 ±0.67[ab] | 0.114 ±0.031[ab] | 304.00 ±24.19[a] | 99.93 |
| | QZ_T1 | 329.21 ±10.70[bc] | 330.56 ±19.87b[c] | 2.56 ±0.78[b] | 0.198 ±0.069[a] | 304 ±8.47[a] | 99.9 |
| | QZ_T2 | 372.24 ±21.36[a] | 370.84 ±19.95[a] | 2.93 ±0.50[ab] | 0.117 ±0.0584[ab] | 343 ±12.56[a] | 99.89 |
| | QZ_T3 | 357.13 ±20.83[ab] | 356.10 ±20.13[abc] | 3.74 ±0.55[a] | 0.060 ±0.036[b] | 350.99 ±22.78[a] | 99.96 |
| | QZ_T4 | 347.05 ±9.26[abc] | 346.70 ±19.74[abc] | 3.30 ±0.61[ab] | 0.085 ±0.0414[b] | 335 ±36.65[a] | 99.94 |
| | QZ_T5 | 365.18 ±21.00[a] | 364.08 ±19.72[ab] | 3.45 ±0.28[ab] | 0.079 ±0.0341[b] | 352 ±39.01[a] | 99.93 |
| Bacteria | QZ_CK | 2,392.68 ±72.35[c] | 2,375.06 ±65.80[c] | 6.24 ±0.77[a] | 0.010 ±0.006[a] | 2,085.00 ±52.40[a] | 98.39 |
| | QZ_T1 | 2,648.72 ±55.88[b] | 2,601.15 ±78.16[b] | 6.50 ±0.11[a] | 0.006 ±0.004[a] | 2,283.00 ±51.23[a] | 98.16 |
| | QZ_T2 | 2,859.54 ±47.52[a] | 2,866.95 ±12.62[a] | 6.58 ±0.64[a] | 0.003 ±0.000[a] | 2,365.99 ±34.44[a] | 97.78 |
| | QZ_T3 | 2,465.07 ±60.62[c] | 2,423.18 ±54.75[c] | 6.47 ±0.30[a] | 0.004 ±0.002[a] | 2,159.99 ±32.92[a] | 98.39 |
| | QZ_T4 | 2,612.38 ±68.10[b] | 2,600.23 ±71.27[b] | 6.56 ±0.82[a] | 0.004 ±0.004[a] | 2,276.00 ±65.95[a] | 98.24 |
| | QZ_T5 | 2,601.38 ±29.18[b] | 2,556.19 ±46.13[b] | 6.49 ±0.91[a] | 0.005 ±0.002[a] | 2,260.00 ±70.18[a] | 98.25 |

**Notes.**
The datas in the table are the mean ± standard error.
Different lowercase letters in dicate significant differences at 0.05 level between different treatments.

the five different fungicide treatments compared to the control treatment. Chao1 richness results showed there were no significant differences in the soil fungal communities in the inter-root soil of aubergine between two of the fungicide treatments, QZ_T2 and QZ_T5, and the control treatment (QZ_CK). The differences in the inter-root soil bacterial community of aubergine between the QZ_T1, QZ_T2, QZ_T4 and QZ_T5 fungicide treatments and the QZ_CK control treatment reached a significant level. Abundance-based coverage estimator (ACE) results showed the differences in the inter-root soil bacterial community between the QZ_T2, QZ_T3 and QZ_T5 fungicide treatments and the QZ_CK control treatment were not significant. Results found the differences between the soil fungal communities in the inter-root area of aubergine treated with three fungicides (QZ_T2, QZ_T3, QZ_T5) and those in the QZ_CK control treatment reached significant levels, and the differences between the soil bacterial communities in the inter-root area of aubergine treated with QZ_T1, QZ_T2, QZ_T4, or QZ_T5, and those in the control group reached significant levels. This may be because fungicide agents affected the community diversity to a certain extent, which was manifested as a decrease in community diversity and an increase in evenness.

## Analysis of the structural composition of soil bacterial and fungal communities after treatment with different microbial agents

As shown in Fig. 3, the relative abundance of microorganisms in the soil of each treatment group changed significantly after the application of different microbial agents. The samples treated with different microbial agents contained a total of 10 fungal phyla, with nine fungal phyla accounting for an average of 90.78% of the fungi across treatments (the tenth fungal phylum was "unclassified fungal phylum"). Figure 3A shows that *Ascomycota* and *Mortierellomycota* were the co-dominant species, accounting for 71.44%–96.88%

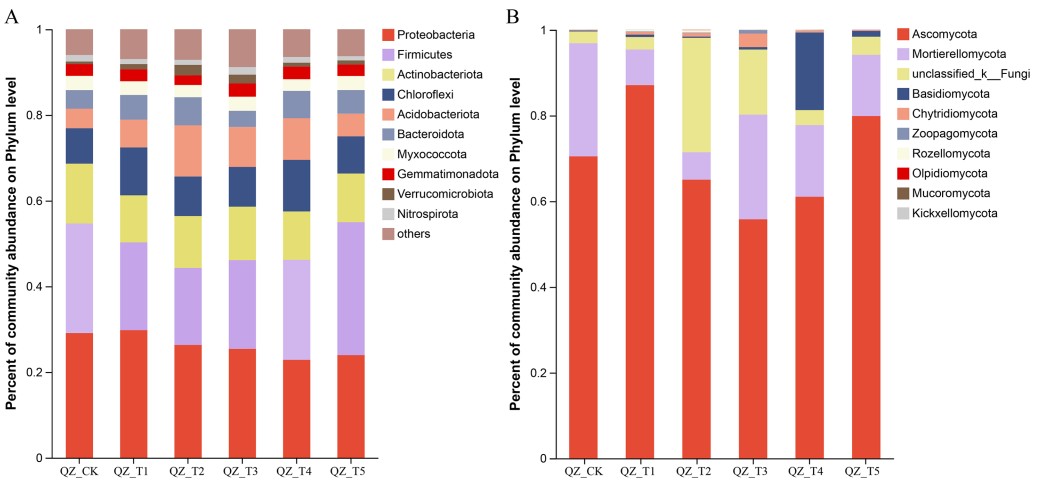

**Figure 3** Relative abundance at phylum level.

of the fungi in samples treated with different microbial agents. The relative abundance of *Ascomycota* increased by 23.59% with the QZ_T1 treatment and by 13.37% with the QZ_T5 treatment; the relative abundance of *Ascomycota* decreased with the remaining three treatments. The relative abundance of *Mortierellomycota* decreased with all five treatments, with the biggest decrease seen in QZ_T2 where *Mortierellomycota* decreased by 75.78%. The relative abundance of *Zoopagomycota* decreased by 81.48% with the QZ_T5 treatment. Except for the QZ_T3 treatment, where the relative abundance of *Rozellomycota* did not change, all treatments showed an increase in the relative abundance of *Rozellomycota*. The relative abundance of *Chytridiomycota* increased by 51.28% in QZ_T5, but increased 3.72–35.54 times with all other microbial treatments. The relative abundance of *Rozellomycota* increased 27.11-fold with the QZ_T2 treatment, *Chytridiomycota* increased 35.54-fold with the, and *Basidiomycota* increased by 93.59 times with the QZ_T4 treatment. The relative abundance of *Olpidiomycota* decreased 1-fold with the QZ_T1 and QZ_T4 treatments and *Mucoromycota* decreased 1-fold with the QZ_T2 treatment; *Olpidiomycota* and *Mucoromycota* decreased with all other treatments. Only QZ_T2 increased the relative abundance of *Kickxellomycota*, with a relative abundance of 0.0089%. All five treatments showed an increase in the relative abundance of the unclassified phyla.

The samples treated with different microbial agents contained a total of 41 bacterial phyla, with the top 10 bacterial phyla accounting for an average of 93.06% of the bacteria across treatments (Fig. 3B). Are these numbers all compared to the control group *Proteobacteria* and *Firmicutes* were the co-dominant phyla, accounting for 44.29%–54.98%. *Firmicutes* increased by 21.26% with the QZ_T5 treatment and *Proteobacteria* increased by 2.37% with the QZ_T1 treatment; *Firmicutes* and *Proteobacteria* decreased with all other treatments, with *Firmicutes* decreasing by 29.87% and *Proteobacteria* decreasing by 9.52% with the QZ_T2 treatment. *Actinobacteriota* decreased after the QZ_T2, QZ_T3, and QZ_T4 treatments (by up to 12.76% with the QZ_T2 treatment), but increased 1.62 times with the QZ_T2 treatment. *Bacteroidota* decreased 12.77% with the QZ_T3 treatment, but

increased with all other treatments (by up to 50.58% with the QZ_T2 treatment). The relative abundance of *Verrucomicrobia* increased with all five treatments (by 2.96 times in QZ_T2). The relative abundance of *Gemmatimonadpta* and *Nitrospirota* decreased by 16.89% and 22.86%, respectively, with the QZ_T2 treatment. The unclassified bacterial phyla increased with all five treatments: by 15.82% with QZ_T1, 18.79% with QZ_T2, 47.39% with QZ_T3, 7.58% with QZ_T4 and 4.42% with QZ_T5, respectively.

A total of 183 species were detected in the genus-level fungal community. As shown in supplement 1A, the top 10 fungal genera accounted for an average of 76.30% across treatments. Species with a relative abundance greater than 1% included *Mortierella*, *Lophotrichus*, *Chaetomium*, *Thermomyces*, *Coprinellus*, *Mycezliophthora*, *Aspergillus*, *Chrysosporium* and *Cladosporium*. All treatments except QZ_T1 decreased the relative abundance of *Lophotrichus*. The relative abundance of *Chaetomium* increased with all five treatments, with the relative abundance increasing by 33.50%–233.09%. Unclassified fungi, *Thermomyces* and *Mycezliophthora* also increased with all five treatments, with the relative abundances increasing by 873.05%, 480.23% and 322.14%, respectively, with the QZ_T2 treatment. The relative abundance of *Coprinellus* decreased by 57.89% with the QZ_T2 treatment, but increased by 158.89%–19,476% with the other four treatments. *Aspergillus* decreased with all five treatments, decreasing the most with the QZ_T2 treatment (70.59%), followed by the QZ_T1 treatment (62.89%), and decreasing the least with the QZ_T3 treatment (29.06%). The relative abundance of *Cladosporium* decreased with all five treatments by 97.15%–99.67%. The relative abundance of *Chrysosporium* increased by 488.13% and 220.73% with the QZ_T3 and QZ_T5 treatments, respectively, and decreased by 44.65%, 65.72% and 12.54%, with the QZ_T1, QZ_T2, and QZ_T4 treatments, respectively. These results indicated that the application of microbial agents largely promoted increases in the relative abundance of *Chaetomium*, *Thermomyces* and *Mycezliophthora* and decreases in the relative abundance of *Aspergillus* and *Cladosporium*; these changes significantly affected soil fungal community composition at the genus level.

Supplemental file 1B shows the top 10 dominant bacterial genera, which together accounted for 16.21% of the total, with six of the ten being undetermined bacterial genera. Compared with the control, only the QZ_T5 treatment increased the relative abundance of *Bacillus* (by 38.88%), with the relative abundance decreasing with the other four treatments.

## Relationship between soil environmental factors and microbial communities

A redundancy analysis (RDA) of soil enzyme activities (urease, sucrase, alkaline phosphatase and catalase) and photosynthetic parameters (net photosynthetic rate, transpiration rate, intercellular carbon dioxide concentration, stomatal conductance and water utilisation) was performed to assess their effects on the structure and composition of microbial communities (Fig. 4). The blue vector in the figure represents the response variable of soil fungal community phylum classification level, and the red vector represents the explanatory variables of soil enzyme activity and photosynthetic parameters; the longer the straight line of the explanatory variables, the greater the influence on the fungal community structure.

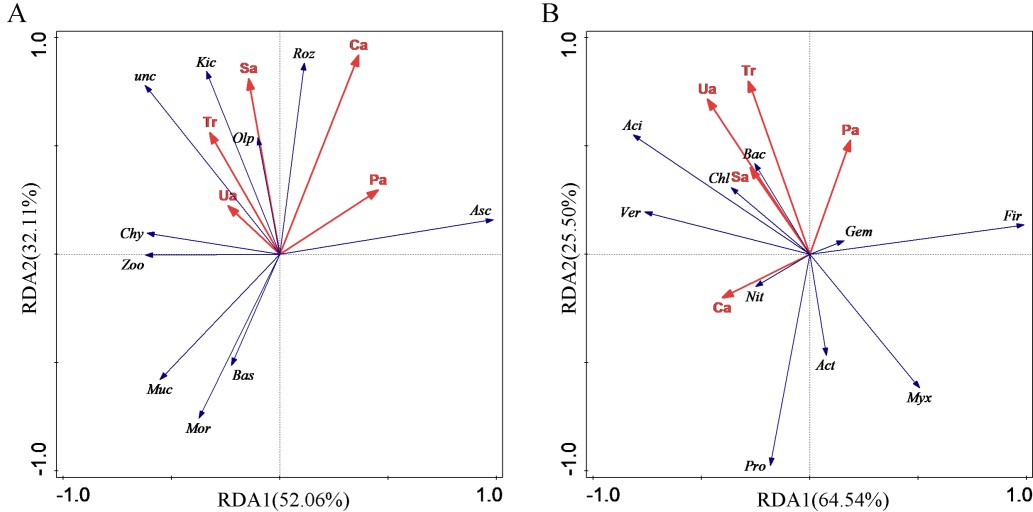

**Figure 4  Redundancy analysis of eggplant.** Note: Tr, Transpiration rate; Sa, sucrase; Pa, alkaline phosphatase; Ca, catalase; Ua, urease. (A) Asc, *Ascomycota*; Mor, *Mortierellomycota*; Zoo, *Zoopagomycota*; Roz, *Rozellomycota*; Bas, *Basidiomycota*; Chy, *Chytridiomycota*; Olp, *Olpidiomycota*; Muc, *Mucoromycota*; Kic, Kickxellomycota; unc, unclassified_k_Fungi; (B) Pro, *Proteobacteria*; Fir, *Firmicutes*; Act, *Actinobacteriota*; Chl, *Chloroflexi*; Aci, *Acidobacteriota*; Bac, *Bacteroidota*; Myx, *Myxococcota*; Gem, *Gemmatimonadota*; Ver, *Verrucomicrobiota*; Nit, *Nitrospirota*.

The correlation between the response variable and the explanatory variables was expressed as the cosine of the angle between the two variables.

The results of the RDA analysis of the phyla with the top 10 relative abundances (the top nine fungal phyla in terms of relative abundance, accounting for a total of 90.78% of the total variance) and the five environmental factors screened by the variance inflation factor are shown in Fig. 4A. It can be seen that the structural features of the fungal communities explained 52.06% and 32.11% of the total variance in the 1st ordering axis (RDA1) and 2nd ordering axis (RDA2), respectively, with a total explanation of 84.17% of the total variance. Fungal community structure on the first axis was positively correlated with Ca and Pa, with Ca explaining 34.2% ($p = 0.158$) and Pa explaining 21.4% ($p = 0.296$) of the variance in fungal community characteristics; it was negatively correlated with Sa, Tr, and Ua, with Ua explaining 18.8% ($p = 0.236$) of the variance in fungal community characteristics and Sa explained 19.3% ($p = 0.288$) of the variation in fungal community characteristics; RDA analysis showed that Ca, Pa, and Sa were the main environmental factors affecting the structural composition of soil fine fungal communities.

The results of the RDA analysis of the selected bacterial dominant phyla with the top 10 relative abundance (93.06% on average) and the five environmental factors screened for variance inflation factors are shown in Fig. 4B. It can be seen that the structural characteristics of the bacterial community explained 64.54% and 25.50% of the total variance in the 1st ordering axis (RDA1) and 2nd ordering axis (RDA2) respectively, which altogether explained 90.04% of the total variance. Bacterial community structure on the first axis was positively correlated with Pa, which explained 23.2% of the variance in bacterial

community characteristics ($p = 0.29$); bacterial community structure on the first axis was negatively correlated with Ca, Ua, Tr, and Sa, where Ca explained 32.1% of the variance in bacterial community characteristics ($p = 0.106$), Ua explained 28.8% of the variance in bacterial community characteristics ($p = 0.204$), and Tr explained 11.2% ($p = 0.262$) of the variation in bacterial community characteristics; RDA analysis showed that Ca, Ua, Pa, and Tr were the main environmental factors affecting the structural composition of soil bacterial communities.

## DISCUSSION

### Effect of fungicide application on photosynthesis in aubergine

Photosynthesis is the main way plants convert light energy into organic matter (*Luo & Keenan, 2020*). The photosynthetic capacity of a plant represents its assimilation capacity and its ability to form carbohydrates (*Zhang et al., 2023*). The net photosynthetic rate reflects the accumulation of organic matter in plants and is an important indicator of the photosynthetic capacity of plants. Stomatal conductance and transpiration rate reflect the water-use efficiency of the vegetation, the photosynthetic efficiency, and the uptake and utilisation of carbon dioxide (*Wang et al., 2021*). Microorganisms can regulate the expression of plant functional genes through a variety of pathways, thereby increasing the chlorophyll content of leaf cells, improving PS II activity, and promoting photosynthesis (*Tanwar et al., 2022*; *Muhammad et al., 2021*). In this study, there were significant differences between the photosynthetic parameters of the plants with microbial agent treatments and those in the control group, indicating that the application of microbial agents is conducive to the enhancement of the photosynthetic characteristics of aubergine leaves. The application of microbial agents of different compositions had different effects, which is in line with the results of the study by *Hamani et al. (2023)*. The differences in net photosynthetic rate, stomatal conductance, intercellular $CO_2$ concentration and transpiration rate between the QZ_T2 and QZ_T5 treatments and the rest of the treatments reached a significant level. This result, in combination with the instantaneous water utilization rate of the two treatments, indicated that microbial fungicides effect the maintenance of water balance in the plant body, the plant's adaptability to the environment, and are beneficial to the photosynthetic characteristics of eggplants. This may be due to the fact that the metabolites produced by microorganisms during fermentation and reproduction, such as *gibberellins* (GA) and *cytokinins* (CTK), as well as other plant growth regulators contained in mycorrhizal fungicides, can directly promote the growth of plants, increase the content of chlorophyll in the plants, prolong the growing period of the plant's root system, extend the contact surface of the root system with the soil, and facilitate the plant's absorption of nutrients necessary for photosynthesis, thus increasing the photosynthetic efficiency of the plant (*Contreras-Cornejo et al., 2022*).

### Relationship between fungicide application and soil enzyme activities of aubergine

Soil enzyme activities are considered to be sensors of soil fertility. They are directly involved in the transformation of substances in the soil as well as the release and fixation of soil

nutrients. The level of soil enzyme activity reflects the activity and biochemical reactions of microorganisms in the soil (*Carlson et al., 2015*). This study showed that the application of different microbial agents had different effects on soil sucrase, alkaline phosphatase, catalase and urease activities. There was a significant difference between the soil sucrase activity with the QZ_T2 treatment, which had the highest soil sucrase activity, and soil sucrase activity with the other treatments. The differences between catalase activities of the QZ_T1 and QZ_T2 treatments and the rest of the treatments reached a significant level. The urease activity of the QZ_T4 treatment was the highest and the enzyme activities of all the microbial treatments were all higher than the control treatment, which is consistent with the findings of *Deng et al. (2021)*. This study also found that alkaline phosphatase in soil has a significant influence on the structure of bacterial communities. Soil is a complex ecosystem and microbial efficacy depends on the microbial population, competitiveness with indigenous microorganisms, as well as the level of soil organic matter content and redox potential (*Qi et al., 2022*; *Qi et al., 2021a*; *Qi et al., 2021b*). The application of microbial agents can improve the activity of enzymes in the inter-root soil, which in turn improves the soil nutrient environment, making the abundance of beneficial microorganisms rise, playing a micro-regulatory role in ecosystems (*Tu et al., 2020*). This may be the potential influence of some environmental variables on the assembly of inter-root microbial communities, which play a key role in the construction of inter-root microbial communities. The deterministic and stochastic selection of communities by environment changes also regulates the construction of inter-root microbial communities (*Chen et al., 2021*).

## Relationship between fungicide application and soil microbial communities in aubergine inter-root soils

The microbial community maintains the ecological balance of the soil and is one of the most important indicators of soil health. Soil microorganisms form a dynamic system of interaction with the root system through their own activities, which in turn affect the crop (*Suneetha, 2020*). Some microorganisms are able to secrete hormonal probiotic substances such as growth hormones and gibberellins, while releasing compounds such as antimicrobials and flavonoids that are involved in the systemic defence response, increasing the vigour of the plant root system (*Chandra, Chandra & Tripathi, 2021*). The root system of the plant then releases secretions that recruit beneficial microorganisms, leading to changes in the inter-root microbial community structure alongside changes in the number of inter-root soil fungi and bacteria (*Afzal et al., 2019*). The analysis of soil microbial community composition in this study revealed that at the phylum level, *Ascomycota* and *Mortierellomycota* were the co-dominant species. The relative abundance of *Basidiomycota*, *Chytridiomycota*, *Zoopagomycota* and *Rozellomycota* all showed an increase after microbial treatments. The majority of taxa in *Ascomycota* are saprophytes, which are important drivers of carbon and nitrogen cycles in agro-ecosystems, and play an important role in soil stability, plant biomass decomposition and other processes (*Challacombe et al., 2019*), whereas *Basidiomycota* are more capable of decomposing and transforming difficult-to-biodegrade compounds than *Ascomycota*. Ascomycetes are

lignin-degrading, and are more abundant in environments with better soil quality (*Li et al., 2019*). In this study, the relative abundance of *Ascomycota* increased after application of microbial agents. Soil *Basidiomycota* abundance increased after the application of microbial fungicides, indicating that the fungicides promote soil quality, likely because they provide a large amount of carbon, nitrogen and other nutrients for the growth and development of *Basidiomycota* (*Ahsan et al., 2023*). *Zoopagomycota* mostly prey on protozoa such as amoebas or nematodes, and can also parasitise them inside and outside the body; *Chytridiomycota* mostly decay on plant and animal remains or parasitise aquatic plants, algae, small animals and other fungi, and a few parasitise on higher seed plants; most species are capable of breaking down cellulose and chitin *Kong et al. (2021)*. At the genus level, the application of microbial agents largely contributed to the increase in the relative abundance of the genera *Chaetomium*, *Thermomyces*, *Mycezliophthora*, *Coprinellus*, and *Chrysosporium* and the decrease in the relative abundance of *Aspergillus* and *Cladosporium*. *Chaetomium* is a larger genus of *Ascomycota* and is an important group of biocontrol fungi that are effective against plant pathogens (*Soytong et al., 2001*). *Ogundeji et al. (2021)* found that enrichment of inter-root *Trichoderma* may be one of the main mechanisms of biochar control of yellow wilt disease in aubergine. *Thermomyces*, a genus of thermophilic fungi, is heat tolerant and produces thermophilic enzymes including cellulase, protease, amylase and lipase (*Kumari et al., 2024*). *Myceliophthora*, a saprophytic, thermophilic mould, is widely present in the environment and is effective at degrading wood and plant biomass and producing heat-resistant enzymes (*Singh, 2016*). *Aspergillus* is a widely distributed fungus, is a common food and environmental pollution bacteria, and some species are important human and animal conditional pathogens (*Schubert et al., 2018*). Studies have shown that *Cladosporium* is a common pathogen that can cause disease in plants (*Yang & Li, 2022*). *Chrysosporium* is a fungal species that is widely distributed in the environment. Khatami found that *Chrysosporium* promotes the humification process by degrading lignin-derived aromatic molecules and generating new aliphatic molecules, which facilitate the accumulation and storage of organic matter in the soil *Khatami et al. (2019)*. *Liu et al. (2022)* found that *Chrysosporium* is a lignin-degrading bacterium with inhibitory effects on plant pathogenic microorganisms, and may promote the growth of clumping mycorrhizal fungi and the production of EE-GRSP through an increase of soil organic carbon. Shang also found that *Chrysosporium* can promote the degradation of insecticides in the soil, and further studies showed that *Chrysosporium* is a class of keratinophilic fungi that can produce keratinase, cellulase, lipase, inulinase and $\alpha$-galactosidase, a variety of enzymes and some secondary metabolites *Shang et al. (2023)*. Many studies have shown that keratinophilic fungi treatment of keratin waste can increase the quick-acting nitrogen in the soil, thereby increasing the effective use of nitrogen by plants (*Passari et al., 2016*), but will also produce tryptophan, which plays an important role in plant growth and development (*Kshetri et al., 2018*). These studies help explain why microbial agents in the present study made the relative abundance of degrading bacteria and biocontrol bacteria in the inter-root soil of aubergine increase and the relative abundance of pathogenic bacteria and pathogenic bacteria decrease.

Plant inter-root bacterial communities are influenced by the plant species and the plant's growth environment. Changes in the plant's growth environment can induce changes in the plant transcriptome and metabolome, selectively alter the abundance or diversity of certain types of inter-root microorganisms by altering the secretion products in the roots, and form a microbial community structure and function more favourable to the growth and development of the host plant to enhance the plant's the survival rate and environmental adaptability (*Liu et al., 2020*). After mycorrhizal treatment, at the phylum level of the soil, *Proteobacteria* and *Firmicutes* were the co-dominant species. The relative abundance of *Actinobacteriota* was slightly reduced, while *Chloroflexi*, *Acidobacteriota*, *Bacteroidota*, *Planctomycetota*, *Nitrospirota*, and *Verrucomicrobia* showed a slight increase in species abundance compared to the control. *Planctomycetota* has the potential for carbon fixation and degradation of organic matter. oxidises ammonia-generated nitrite, through the nitrifying bacteria in the soil, to produce the crop-ready form of nitrate, which is a nitrogen source, and *Nitrospira* can indirectly increase soil nitrogen fertility and play a key role in soil nitrogen metabolism (*Daims & Wagner, 2018*). *Verrucomicrobia* has a variety of sparsely linked non-ribosomal peptide synthetase systems, which are associated with a variety of material cycles (*Crits-Christoph et al., 2018*). These cycles utilise complex carbohydrates and may also be fully involved in the methane cycle, influencing the dynamic balance of the carbon and nitrogen cycles (*Khadem et al., 2010*). *Cyanobacteria* had the highest increase in species abundance after microbial agent application. *Cyanobacteria* is an ancient autotrophic bacterium with strong adaptability to extreme environments (*Farrokh et al., 2019*); it improves soil physicochemical properties and the biological control of plant pathogens (*Prasanna et al., 2011*), which can provide a more suitable microenvironment for plant enrichment of essential elements for growth. *Kalam et al. (2020)* found that *Acidobacteriota* is one of the most important microorganisms in the soil flora because it participates in several important material cycling processes in the soil. Because *Acidobacteriota* is involved in the degradation of plant residue polymers in the soil and participates in the carbon metabolism, iron cycling, and photosynthesis, its increased abundance is conducive to the promotion of the material and nutrient cycling of soil, which is of great importance in ecosystems (*Coluccia & Besaury, 2023*). *Bacteroidota* contains a number of genera capable of secreting phosphatases and organic acids, which can promote the conversion of soil insoluble phosphorus and increase phosphorus effectiveness (*Fraser et al., 2015*). *Bacteroidota* also participates in the degradation of inter-root high molecular weight organic matter, with a very strong nutrient metabolism, and can convert high molecular compounds, such as proteins, into small molecular compounds (*Zhang et al., 2011*; *Wolińska et al., 2017*). *Bacteroidota* also excels in degrading complex organic matter in the biosphere, which is often regarded as a biological indicator of the adequacy of soil utilisation efficiency in agriculture, while *Chloroflexi* break down polysaccharide substances within the soil into organic acids and hydrogen, facilitating the degradation of organic matter within the soil. *Chloroflexi* also have a positive effect on cellulose degradation (*Zhang et al., 2020*), play an important role in carbon and nitrogen metabolism (*Hou et al., 2018*), and are an important group of bacteria that promote denitrification (*Chen et al., 2018*). At the genus level, the use of microbial agents promotes *Bacillus*, *Paenisporosarcina*,

the unnamed taxonomic genera of the order *Actinomarinales*, the unnamed taxonomic genera of the family *Geminicoccaceae* and the unnamed taxonomic genera of the order *Alphaproteobacteria*, the relative abundance of unnamed genera of the order *Bacteriap* 25 decreased and the relative abundance of unnamed genera of the family A4b and unnamed genera of the order *Vicinamibacterales*, *Sphingomonas*, and other genera increased. Members of the genus *Sphingomonas* have been reported to have a variety of ecological functions, such as the degradation of complex organic matter, antagonism of phytopathogenic fungi, as well as the secretion of extracellular polysaccharides, and the promotion of plant uptake (*Mazoyon et al., 2023*). The increase in the number of important beneficial functional bacterial taxa in the bacterial community seen in the inter-root soil after bacterial treatment allows the timely decomposition of proteins and other substances in the soil, improving the quality of the soil, providing crops with required nutrients, promoting crop growth, and reducing the incidence of soil diseases (*Qi et al., 2021a*; *Qi et al., 2021b*).

The effect and sustainability of microbial fungicide use in the field are affected by the type of microbial fungicide and effective bacterial concentration used, the soil characteristics, planting pattern, different varieties of aubergine, and technology used. Currently, there is a complex variety of production types of single or complex microbial agents. The synergistic effect of multiple microbial applications in soil is also an important scientific issue. In this study, only five kinds of single microbial fungicides were used with 'Big Red Robe' aubergines in a soil field test. Further research is needed to see whether microbial fungicides have the same effect on other varieties of aubergine. The persistence of microbial fungicides in soil and the relationship between the role of microbial fungicides and organic and chemical fertilisers in the growth and quality improvement of plant fruits were not addressed in this paper and will continue to be explored in subsequent trials.

## CONCLUSIONS

This study showed that the application of different microbial agents had different degrees of ameliorative effects on photosynthesis and soil enzyme activities in aubergine, mainly enhancing the gas exchange parameters such as the net photosynthetic rate, stomatal conductance and transpiration rate, as well as enhancing soil enzyme activities. The composition of the microbial community in the inter-root soil of aubergine changed with the application of mycorrhizal agents, with an increase in the relative abundance of beneficial genera and a decrease in the relative abundance of pathogenic genera. These changes affect the interaction of the microbial community network, community stability and the community's resistance to environmental changes. Therefore, microbial fungicides may help maintain the stability of soil microbiota in the inter-root zone of aubergine and may have a mitigating effect on continuous cropping disorder. The results of this study provide a multidirectional demonstration for understanding the relationship between microbial agents and the diversity of the inter-root microbial community of aubergine. The results of this study help elucidate the microbial community construction process and the potential mechanism of different microbial agents on the inter-root soil of aubergine. This study also provides a theoretical basis for the development of microbial

agents, especially composite microbial agents, for soil improvement and for inter-root soil microbiome stabilisation.

## ACKNOWLEDGEMENTS

Thanks to PeerJ and to everyone who helped with the experiment and the completion of the manuscript.

### Funding

This work was funded by Youth Genetics. The funders had no role in study design, data collection and analysis, decision to publish, or preparation of the manuscript.

### Grant Disclosures

The following grant information was disclosed by the authors:
Youth Genetics.

### Competing Interests

The authors declare there are no competing interests.

### Author Contributions

- Longxue Wei conceived and designed the experiments, performed the experiments, analyzed the data, prepared figures and/or tables, and approved the final draft.
- Jinying Zhu conceived and designed the experiments, performed the experiments, authored or reviewed drafts of the article, and approved the final draft.
- Dongbo Zhao analyzed the data, prepared figures and/or tables, and approved the final draft.
- Yanting Pei performed the experiments, authored or reviewed drafts of the article, and approved the final draft.
- Lianghai Guo analyzed the data, authored or reviewed drafts of the article, and approved the final draft.
- Jianjun Guo analyzed the data, authored or reviewed drafts of the article, and approved the final draft.
- Zhihui Guo performed the experiments, prepared figures and/or tables, field management and documentation, and approved the final draft.
- Huini Cui performed the experiments, prepared figures and/or tables, field management and documentation, and approved the final draft.
- Yongjun Li analyzed the data, authored or reviewed drafts of the article, field management and documentation, and approved the final draft.
- Jiansheng Gao analyzed the data, authored or reviewed drafts of the article, and approved the final draft.

### DNA Deposition

The following information was supplied regarding the deposition of DNA sequences:

The sequences are available at NCBI BioSample: SAMN39824882, SAMN39824883, SAMN39824884, SAMN39824885, SAMN39824886, SAMN39824887.

### Data Availability

The raw data are available in the Supplemental Files.

### Supplemental Information

Supplemental information for this article can be found online at http://dx.doi.org/10.7717/peerj.17620#supplemental-information.

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
