# Peer review of "Microbial fungicides can positively affect aubergine photosynthetic properties, soil enzyme activity and microbial community structure"

_PeerJ, doi:10.7717/peerj.17620_

## Round 0.1 · original submission · Major Revisions

While the current research topic is interesting, the narrative flow and data presentation could be improved. The authors should focus their discussion more directly on their findings.

Additionally, biochemical assays may be necessary to support their assumptions, as the presence of taxa indicated by 16SRNA gene and ITS sequences does not necessarily guarantee functional activity.

**Language Note:** The review process has identified that the English language must be improved. PeerJ can provide language editing services - please contact us at [email protected] for pricing (be sure to provide your manuscript number and title). Alternatively, you should make your own arrangements to improve the language quality and provide details in your response letter. – PeerJ Staff

·

Basic reporting

Microbial fungicides can positively affect eggplant soil enzyme activity and microbial community structure and function (#95665)
Wei and Gao et al., presented a very interesting topic of where they studied the effect of microbial agents on the enzyme activities and microbial community construction and potential functions of inter-root soil of Aubergine / Eggplant or Brinjal. They also shade light on adaptability of inter-root microbes to the environmental factors so as to provide theoretical basis for stability of the microbiology of inter-root soil of aubergine and ecological preservation of farmland soil.
It is an interesting topic however their selection of aubergine is kind of complex to understand. It’s not clear why they chose aubergine as opposed to any other crop/plan for that matter. Even though the topic is interesting and authors have performed High throughput sequences they haven’t done a great job representing the data. If the data is presented in clear and concise manner it will be a nice paper and might help wider audience interested in aubergine and soil communities affecting the growth.
Despite these merits this manuscript has a tonne of flaws and needs to be seriously re-written and including the figures and data representation.

Experimental design

Some of the highlights are mentioned below,
Line 18: Sentences in the background on Line 18 should be broken down into a couple of three sentences to make it easier.
All the bacterial names should be italicized or underlined.
Full forms should be used whenever required and at least once it appears for the first time in the text. For example, Line 29 what is ITS high throughput sequencing?
Line 64, pressure to control pest maybe requires rewriting or paraphrasing.
Material and methods:
Pay special attention to abbreviations, symbols and units throughout the material and methods section or throughout the manuscript for that matter.
Line 169: Difficult to understand where sentences start and ends or what?
Line 181: what /who is Meggie?
Line 204: LSD?
Line 209: Pay attention to writing, grammar and punctuation, please.
Line 234: where does this sentence come from? What is OTU?
Line 246: Please elaborate in brief what re Sobs, Shannon and Chao are?
Line 259: Please double check sentences.
Line 781 AND 782: The acknowledgement is very funny!
Table 1 and Table 2 : Please rewrite the whole thing!
Figure 1: Figure legends are missing, and data should be reported on the figures. Please mention biological and technical replicates, N, and statistical significances. Please check the x and y-axes and nomenclature. Same goes with Figure 2, in fact all of them!
It is highly recommended to double check the grammar, spellings, and comprehension.

Validity of the findings

Microbial fungicides can positively affect eggplant soil enzyme activity and microbial community structure and function (#95665)
Wei and Gao et al., presented a very interesting topic of where they studied the effect of microbial agents on the enzyme activities and microbial community construction and potential functions of inter-root soil of Aubergine / Eggplant or Brinjal. They also shade light on adaptability of inter-root microbes to the environmental factors so as to provide theoretical basis for stability of the microbiology of inter-root soil of aubergine and ecological preservation of farmland soil.
It is an interesting topic however their selection of aubergine is kind of complex to understand. It’s not clear why they chose aubergine as opposed to any other crop/plan for that matter. Even though the topic is interesting and authors have performed High throughput sequences they haven’t done a great job representing the data. If the data is presented in clear and concise manner it will be a nice paper and might help wider audience interested in aubergine and soil communities affecting the growth.
Despite these merits this manuscript has a tonne of flaws and needs to be seriously re-written and including the figures and data representation.

Reviewer 2 ·

Basic reporting

Language needs to be improved greatly throughout the manuscript. Please use professional service to polish the language.

Species is taxonomy level as well as phylum and genus. Species couldn’t be used to represent phylum or genus. Please correct it throughout the manuscript.

In the Introduction, please explain why the five treatments were selected in this study and what are the potential impacts of these five microbial agents. Please also include summaries on the reported progress of their impacts on soil enzymes and root soil microbial communities.

Only very few bacteria have specific functions. Most of 16S rRNA gene or ITS couldn’t be connected to specific functions. Please delete the parts about microbial function predictions, unless functional genes or metagenomes were determined in the study.

The study is to evaluate the impact of microbial agents on plant growth, soil enzyme activities and root soil microbial communities. Please determine 1) the statistical significance of difference between control vs each treatment (e.g. Table 2, Fig 1, Table 3 and the corresponding parts in the text. Evaluating significance of difference between treatments is not necessary); 2) if the treatment showing significant difference from control has positive or negative impact, which treatment has the highest positive or negative impact and why. Please use ANOSIM to evaluate the significance of difference in microbial communities between control vs each treatment at the phylum (Fig 3A, 3B), genus (Fig 4A, 4B) and OTU level.

Figs 4A and 4B showed the top 11 genera in the microbial communities. Please indicate what percentages these 11 genera account for in the whole community.

Heatmap in either Fig 5A or 5B didn’t show significant difference between control vs each treatment. For example, Fig 5A showed that the control was closer to T1, T4 and T5 whereas Fig 5B showed that the control was closer to T5. Please correct the corresponding part in the text. The clustering analysis in Fig 5B only included the top 20 bacteria phyla. Please indicate what percentages these 20 taxa represent in the whole community.

The RDA analysis in Fig 6A only included the top 10 fungal phyla. Please include all fungal phyla in the analysis and only visualize the top 10 phyla. Please also include the photosynthetic characteristics (from Table 1) as the explanatory factor in the analysis. Please check the p-value for each explanatory factor and only visualize the explanatory factors which are statistically significant (p<=0.05).

Experimental design

Experimental design is well based on the research question.

Validity of the findings

Please see comments in Basic Reporting.

---

## Round 0.2 · Minor Revisions

Please kindly review the information and ensure the statistical analysis is used correctly.

·

Basic reporting

After carefully looking at the rebuttals and tracked charges I think this manuscript is in a way better position for publication.
Thank you,
Nitin

Experimental design

After carefully looking at the rebuttals and tracked charges I think this manuscript is in a way better position for publication.
Thank you,
Nitin

Validity of the findings

After carefully looking at the rebuttals and tracked charges I think this manuscript is in a way better position for publication.
Thank you,
Nitin

Additional comments

After carefully looking at the rebuttals and tracked charges I think this manuscript is in a way better position for publication.
Thank you,
Nitin

Reviewer 2 ·

Basic reporting

The top ten bacterial genera shown in Fig 4b only accounted for 16.21% of the total bacterial community, which couldn’t represent the bacterial community pattern between treatments and control. The whole manuscript is focused on the phylum level. I would suggest to move Fig 4 to supplement unless there is better reason to include it in the main figures.

The clustering of the microbial communities at the phylum level in Fig 5 didn’t show that the control was separated well from treatments. Please remove Fig 5.

For the RDA analysis in Fig 6, please check the output results from the software. There should be a p value for each explanatory variable. Please focus on significant explanatory variables (p<=0.05). If not, please mainly discuss on the longest vectors of explanatory variables.

Lines 542-546, lines 587-589, lines 591-592: add references

Experimental design

No comment

Validity of the findings

No comment

---

## Round 0.3 · accepted · Accept

Thank you for your revisions, which were acceptable to the reviewer.

Reviewer 2 ·

Basic reporting

The authors addressed my concerns very well. I endorse the publication of this manuscript.

Experimental design

No comment

Validity of the findings

No comment